# Risk factors for death in Welsh infants with a congenital anomaly

Peter S Y Ho [ID],[1] Maria Quigley,[1,2] David F Tucker,[3] Jenny Kurinczuk[1,2]

[1]National Perinatal Epidemiology Unit, Nuffield Department of Population Health, University of Oxford, Oxford, UK
[2]National Institute for Health Research (NIHR) Policy Research Unit - Maternal Health and Care, National Perinatal Epidemiology Unit, Nuffield Department of Population Health, University of Oxford, Oxford, UK
[3]Public Health Wales, Public Health Knowledge & Research, Congenital Anomaly Register & information Service for Wales, Swansea, UK

**Correspondence to**
Dr Peter S Y Ho; Peterhosy@hotmail.com

## ABSTRACT

**Objectives** To investigate risk factors associated with death of infants with a congenital anomaly in Wales, UK.
**Design** A population-based cohort study.
**Setting** Data from the Welsh Congenital Anomaly Register and Information Service (CARIS) linked to live births and deaths from the Office for National Statistics.
**Patients** All live births between 1998 and 2016 with a diagnosis of a congenital anomaly, which was defined as a structural, metabolic, endocrine or genetic defect, as well as rare disease of hereditary origin.
**Main outcome measures** Adjusted ORs (aORs) were estimated for socio-demographic, maternal, infant and intervention factors associated with death in infancy, using logistic regression for all, isolated, multiple and cardiovascular anomalies.
**Results** 30 424 live births affected by congenital anomalies were identified, including 1044 infants who died by the age of 1 year (infant mortality rate: 16.5 per 10 000 live births, case fatality: 3.4%, 30.3% of all infant deaths). Risk factors for infant death were non-white versus white ethnicity (aOR: 2.25; 95% CI: 1.77–2.86); parous versus nulliparous (aOR: 1.24; 95% CI: 1.08–1.41); smoking during pregnancy versus non-smokers/ex-smokers (aOR: 1.20; 95% CI: 1.02–1.40); preterm versus term birth (aOR: 4.38; 95% CI 3.86–4.98); female versus male infants (aOR: 1.28; 95% CI: 1.13–1.46) and the earlier years of the birth cohort (aOR: 0.96; 95% CI: 0.95–0.98 per yearly increase). Infants with a cardiovascular anomaly who received surgery had a lower odds of death than those who did not (aOR: 0.34; 95% CI: 0.15–0.75). Preterm birth was a significant factor for death for all anomalies but the effect of the other characteristics varied according to anomaly group.
**Conclusions** Nearly a third of all infant deaths had an associated anomaly. Improving access to prenatal care, smoking cessation advice, optimising care for preterm infants and surgery may help lower the risk of infant death.

## INTRODUCTION

Congenital anomalies are structural, chromosomal or metabolic abnormalities that occur during intrauterine development[1]; they are the second leading cause of infant death in the UK, accounting for over one-third of all infant deaths.[2 3] To date, population-based studies of risk factors for mortality of infants with congenital anomaly have been limited, with most of the existing studies focusing on a few major anomaly subgroups such as neural tube defects,[4 5] and certain cardiovascular and digestive system anomalies.[6 7]

Previous studies have shown that socio-demographic, maternal, infant and interventional factors can influence the survival of infants born with specific anomalies. For example, maternal black ethnicity, preterm birth, cervicothoracic lesion level of a spina bifida and multiple defects were significantly associated with an increased risk of excess infant deaths in those with neural tube defects.[4 5] Further research on mortality risk factors associated with a wider range of congenital anomalies is needed to inform planning of healthcare and social interventions aimed at reducing infant deaths.

We aimed to investigate risk factors for infant death of infants born with congenital anomalies.

## METHODS
### Study design

A population-based cohort study was conducted using registry data from the Congenital Anomaly Register and Information Service (CARIS) for Wales, linked to births and deaths registration data from the Office for National Statistics (ONS)[3 8] and de-identified for analysis.

### What is known about the subject?

► Congenital anomalies are a leading cause of infant death.
► Evidence about the risk factors contributing to the death of infants with congenital anomalies is limited.

### What this study adds?

► A third of infant deaths in Wales involved an infant with a congenital anomaly.
► Preterm birth was the strongest risk factor for excess infant deaths.
► Socioeconomic factors, including maternal ethnicity and smoking, are risk factors for excess infant deaths.

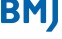

## Study population

The inclusion criteria for this study were all live births between 1998 and 2016 with birth weight ≥500 g, gestational age ≥22$^{+0}$ weeks and a confirmed or probable diagnosis of a congenital anomaly reported to CARIS; these infants were followed up for 1 year after birth.

Congenital anomalies are defined by CARIS as structural, metabolic, endocrine or genetic defects, as well as rare diseases of hereditary origin present in the child or fetus at the end of pregnancy, even if not detected until after birth.[8] All congenital anomalies reported to CARIS are coded using the Royal College of Paediatrics and Child Health adaptation of the 10th revision of the International Classification of Diseases and Related Health Problems (ICD-10 RCPCH). Therefore, ICD-10 'Q', 'P35' and 'P37' codes, as well as other non-'Q' ICD-10 codes of congenital anomalies and rare diseases were included in this study.

## Categorisation of anomaly

Congenital anomalies were categorised according to the European Surveillance of Congenital Anomalies (EUROCAT) subgroup classification[9]; other rare diseases which are not included in the EUROCAT subgroup classification were categorised according to the ICD chapter headings[10] (online supplemental table 1).

Infants were considered as having an isolated congenital anomaly (or disease) if a single anomaly (or disease) was diagnosed and reported to CARIS. Infants were considered as having multiple anomalies (or diseases) if more than one anomaly (or diseases) was diagnosed, either within the same body system or involving different body systems, for example, an infant was diagnosed with a spina bifida and a ventricular septal defect. Infants who were diagnosed with a syndrome involving more than one anomaly (or disease) were considered as having multiple anomalies (or diseases), for example, an infant was diagnosed with a Down syndrome and a congenital heart defect.

## Variables

To informally assess the validity of data (online supplemental table 2), CARIS compares data collected by midwives at booking with that collected by fetal cardiologists who also take a history; some discrepancies are noted, for example, maternal smoking and the more complete data are retained by CARIS. Severity of cardiovascular anomalies is based on the EUROCAT classification.[9]

## Statistical analysis

Adjusted ORs (aORs) were estimated for socio-demographic, maternal, infant and intervention factors associated with infant death, that is, death which occurred in the first year after birth. The analysis was performed separately for: infants with any anomaly; those with isolated anomalies; those with multiple anomalies and those with cardiovascular anomalies.

Infant mortality rate (IMR) was calculated as the number of infant deaths per 10 000 live births, where the baby dies before their first birthday. ONS denominator

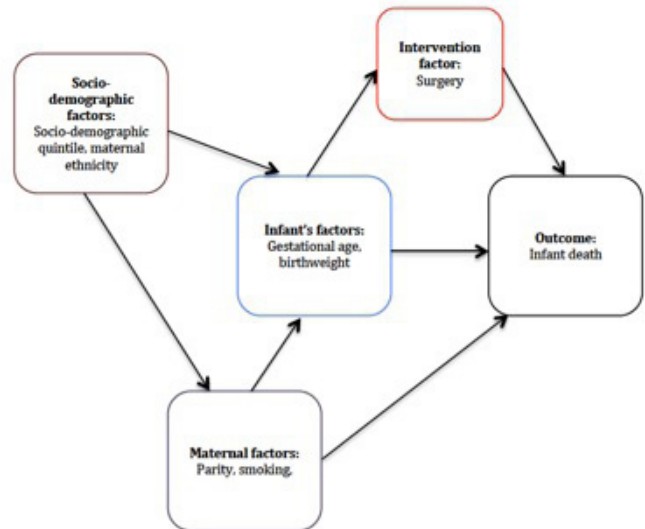

**Figure 1** Choices of the included confounding factors associated with infant death using a directed acyclic diagram.

data for IMR was the total number of live births in Wales between 1998 and 2016. The modelling strategy aimed to identify the strongest risk factors for infant death. The risk factors explored were socio-demographic, maternal, infant and intervention factors (figure 1). A priori factors included maternal ethnicity, infant sex and gestational age at birth, which were selected based on evidence in the literature that they were important socio-demographic and clinical determinants of infant deaths,[4 5 11 12] and hence these variables were included in the final model regardless of their statistical significance in the univariable analysis. For the remaining variables, unadjusted ORs (uORs) were estimated in an univariable analysis to identify variables associated with death (p<0.1), which were then explored in a multivariable logistic regression model to generate aORs. Variables were dropped from the multivariable model using a backward stepwise approach if they did not significantly improve the fit of the data (ie, p<0.05 in the likelihood ratio test) with examination of the results as each variable was removed. Interaction was tested between gestational age and other variables in the final model, p<0.05 was considered as statistically significant.

A sensitivity analysis was conducted for each subgroup by adding year of birth to the final model to investigate if there were any significant changes in the effect of key risk factors associated with infant death. In addition, for variables, including maternal ethnicity, maternal smoking, history of anomalies in previous pregnancy and infant surgery status, the amount of missing data was high (ie, 15%–41%) and associated with other variables (ie, not missing completely at random). As it was not possible to exclude that the missingness may also have been at random, multiple imputation was justified and performed for these variables using multiple imputation chained equations. Sensitivity analyses were conducted

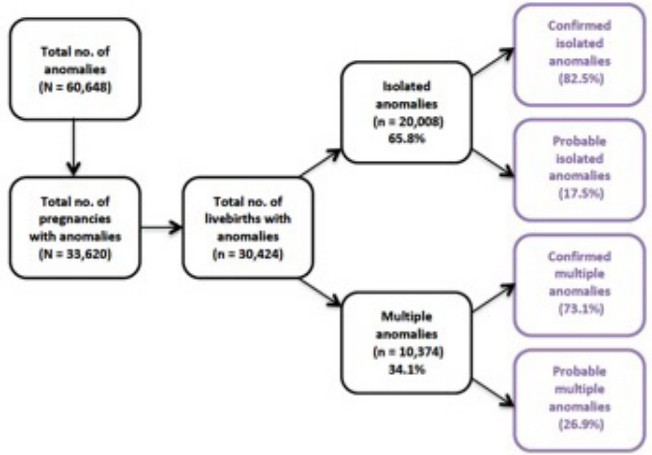

**Figure 2** Anomaly status and categories included in this study: births in Wales from 1998 to 2016. Note: the number of isolated and multiple anomalies does not sum to all anomalies due to coding issues of <0.1% of cases.

using complete case analysis and imputed data to check the robustness of results in the final models. All statistical analyses were performed using Stata V.13.[13]

### Research ethics and statistical disclosure

The de-identified health data for this study was provided by the Secure Anonymised Information Linkage (SAIL) databank; research ethics committee approval was not required for this secondary data analysis study. This study was approved by the SAIL Information Governance Review Panel (IGRP). All statistical disclosure control and checks were strictly followed according to the SAIL Databank and ONS guidelines, to ensure that individual infants were not deductively identifiable.

### Patient and public involvement

All proposals to use data within the SAIL Databank are subject to review by an independent IGRP for privacy risk, data governance and public benefit assessment. The IGRP is made up of a range of independent experts as well as members of the public.

## RESULTS

A total of 632 945 live births occurred to residents in Wales between 1998 and 2016. In this period, 30 424 infants affected by congenital anomalies were identified, of which 20 008 (65.8%) had isolated anomalies and 10 374 (34.1%) had multiple anomalies (figure 2). There were 1044 deaths of infants who were affected by either an isolated anomaly or multiple anomalies; this represented an IMR of 16.5 deaths per 10 000 live births and case fatality of 3.4% for infants with any anomaly. Infants with an anomaly who died represented almost one-third (30.3%) of all infant deaths (n=3443) in this period; about two-thirds (20.7%) of the anomaly related deaths were of infants with multiple anomalies and one-third (9.5%) were of infants with isolated anomalies (table 1). Among all the isolated anomaly subgroups, cardiovascular anomalies were associated with the largest number of infant deaths (n=84), representing an IMR of 1.33 deaths per 10 000 live births and case fatality of 2.7%.

Table 2 shows the characteristics of the infants with any anomalies who died compared with those who survived infancy. Compared with infants who survived, those who died were more likely to live in the most deprived areas (31% vs 28%), be girls (46% vs 39%), have a low birth weight (51% vs 15%), were born preterm (46% vs 15%), to have not received surgery (45% vs 40%), or had mothers who were of non-White ethnicity (9% vs 3%), multiparous (58% vs 50%), smokers (25% vs 17%), had a multiple pregnancy (11% vs 5%) and a maternal history of an anomaly in a previous pregnancy (11% vs 8%).

Table 3 shows the uORs for variables associated with infant death (p<0.1) by different anomaly subgroups. Birth weight and gestational age at birth were highly collinear (correlation coefficient, r=0.75). Only gestational age was, therefore, included in the final model, as it has greater clinical utility.

Table 4 shows that, for infants with any anomaly, significant increases in the adjusted odds of infant death were found among infants born to mothers who were of non-white ethnicity (aOR: 2.25; 95% CI: 1.77–2.86); parous (aOR: 1.24; 95% CI: 1.08–1.41); active smokers during pregnancy (aOR: 1.20; 95% CI: 1.02–1.40); where the infant was born preterm

**Table 1** IMR by congenital anomaly subgroup in Wales (1998–2016 birth cohort)

| Anomaly subgroup* | Live births N=632 945 (%)† | Deaths (%) N=3443 (%)* | Case fatality | IMR per 10 000 live births |
|---|---|---|---|---|
| All anomalies‡ | 30 424 (4.8) | 1044 (30.3) | 3.4% | 16.5 |
| All isolated anomalies | 20 008 (3.2) | 327 (9.5) | 1.6% | 5.17 |
| All multiple anomalies | 10 374 (1.6) | 714 (20.7) | 6.9% | 11.3 |
| Isolated cardiovascular anomalies | 3149 (0.5) | 84 (2.4) | 2.7% | 1.33 |

N represents total observation.
*Total number of deaths in Wales between 1998 and 2016 (ONS and CARIS data) as the denominator.
†Total number of live births in Wales between 1998 and 2016 (ONS data) as the denominator.
‡3The number of isolated and multiple anomalies (reported in CARIS) does not sum to all anomalies due to coding issues of <0.1% of cases.
CARIS, Congenital Anomaly Register and Information Service; IMR, Infant mortality rate; ONS, Office for National Statistics.

**Table 2** Descriptive characteristics of infants with any anomaly who died and those who survived in the first year after birth

| | Died in infancy, n (%) | Survival infancy, n (%) |
|---|---|---|
| Socio-demographic factor | | |
| Townsend quintile | | |
| 1 (least deprived) | 122 (12) | 4564 (16) |
| 2 | 160 (15) | 4594 (16) |
| 3 | 207 (20) | 5361 (19) |
| 4 | 225 (22) | 5861 (21) |
| 5 (most deprived) | 327 (31) | 7886 (28) |
| Maternal ethnicity | | |
| White | 715 (69) | 15807 (56) |
| Other | 89 (9) | 902 (3) |
| Not known/missing | 240 (23) | 11646 (41) |
| Maternal age at birth (years) (n=29400) | | |
| Mean±SD (years) | 28.3±6.5 | 28.2±6.2 |
| ≤24 | 318 (31) | 8514 (30) |
| 25–29 | 288 (28) | 7835 (28) |
| 30–34 | 239 (23) | 7129 (25) |
| ≥35 | 199 (19) | 4849 (17) |
| Maternal factors | | |
| Parity | | |
| Nulliparous | 385 (37) | 11115 (39) |
| ≥1 | 606 (58) | 14116 (50) |
| Multiple pregnancy | | |
| Yes | 114 (11) | 1334 (5) |
| No | 930 (89) | 27022 (95) |
| Maternal smoking | | |
| Smoker | 261 (25) | 4712 (17) |
| Non-smoker/ex-smoker | 541 (52) | 12110 (42) |
| Not known/missing | 242 (23) | 11534 (41) |
| Anomalies in previous pregnancies | | |
| Yes | 117 (11) | 2257 (8) |
| No | 724 (69) | 16153 (57) |
| Not known/missing | 203 (19) | 9946 (35) |
| Infant factors | | |
| Infant sex | | |
| Male | 566 (54) | 17810 (61) |
| Female | 474 (46) | 11550 (39) |
| Birth weight (g) | | |
| Median (IQR) | 2475 (1650–3150) | 3250 (2780–3650) |
| <2500 (low birth weight) | 529 (51) | 4505 (15) |

Continued

**Table 2** Continued

| | Died in infancy, n (%) | Survival infancy, n (%) |
|---|---|---|
| ≥2500 | 510 (49) | 23835 (81) |
| Gestational age (weeks) | | |
| Median (IQR) | 37 (32–39) | 39 (38–40) |
| < $37^{+0}$ (preterm) | 480 (46) | 4510 (15) |
| ≥ $37^{+0}$ (term) | 562 (54) | 23935 (82) |
| Intervention factors | | |
| Surgery | | |
| Performed (or expected) in the first year after birth | 216 (21) | 6405 (22) |
| Not performed or required in the first year after birth* | 465 (45) | 11833 (40) |
| Not known/missing | 363 (35) | 11142 (38) |

Category with small number is not shown due to risk of statistical disclosure.
Note: confirmed and probable cases were included in the analysis.
N represents observations in sub categories.
*Including a minority of conditions which were too severe for surgery.

(aOR: 4.38; 95% CI: 3.86–4.98) and was a girl (aOR: 1.28; 95% CI: 1.13–1.46). Infants who required surgery in the first year after birth had a lower odds of infant death than those who did not (aOR: 0.80; 95% CI: 0.68–0.95). A similar pattern of aORs was seen for isolated anomalies (except that infant sex was not statistically significant) and multiple anomalies (except maternal smoking was not statistically significant). For cardiovascular anomalies, the main risk factors were preterm birth and severity (having a most severe congenital heart defect (CHD vs less CHD: aOR: 229; 95% CI: 90.8–579). Having surgery had a protective effect (aOR: 0.34; 95% CI: 0.15–0.75) for infants with a cardiovascular anomaly. The effects of these variables were not materially different in the sensitivity analyses when adjusted for year of birth (online supplemental table 3) and using complete case analysis and imputation (data not shown). In addition, a more recent birth year was generally associated with a lower odds of death in infants with any congenital anomalies (aOR: 0.96; 95% CI: 0.95–0.98 per yearly increase). There was no significant interaction between gestational age and other variables in the final model.

## DISCUSSION

This population-based cohort study used linked de-identified data from CARIS and ONS to investigate risk factors contributing to an excess risk of death of infants with congenital anomalies born to residents in Wales between 1998 and 2016. We found that infants with any congenital anomaly who died before their first birthday were more likely to be girls, be born preterm, in the earlier years of

**Table 3** uORs for infant mortality by anomaly subgroup

| Factors | All anomalies | Isolated anomalies | Multiple anomalies | Cardiovascular anomalies |
|---|---|---|---|---|
| **Townsend quintile** | | | | |
| 1 (least deprived) | 1 (reference) | 1 (reference) | 1 (reference) | 1 (reference) |
| 2 | 1.30 (1.03–1.65) | 1.30 (0.86–1.95) | 1.29 (0.96–1.73) | 1.61 (0.76–3.41) |
| 3 | 1.44 (1.15–1.81) | 1.27 (0.85–1.89) | 1.48 (1.12–1.96) | 1.25 (0.59–2.66) |
| 4 | 1.44 (1.15–1.80) | 1.35 (0.92–2.00) | 1.37 (1.04–1.80) | 0.91 (0.41–2.01) |
| 5 (most deprived) | 1.55 (1.26–1.92) | 1.46 (1.02–2.11) | 1.56 (1.20–2.02) | 1.33 (0.67–2.66) |
| **Maternal ethnicity*** | | | | |
| White | 1 (reference) | 1 (reference) | 1 (reference) | 1 (reference) |
| Other | 2.18 (1.73–2.75) | 1.77 (1.11–2.82) | 2.34 (1.78–3.08) | 1.00 (0.31–3.26) |
| Not known/missing | 0.46 (0.39–0.53) | 0.59 (0.47–0.75) | 0.56 (0.46–0.68) | 0.71 (0.44–1.13) |
| **Maternal age at birth (years)** | | | | |
| 25–29 | 1 (reference) | 1 (reference) | 1 (reference) | 1 (reference) |
| ≤24 | 1.02 (0.86–1.20) | 1.04 (0.79–1.37) | 0.97 (0.79–1.19) | 1.12 (0.64–1.96) |
| 30–34 | 0.91 (0.77–1.09) | 0.77 (0.56–1.05) | 0.96 (0.78–1.19) | 0.77 (0.41–1.45) |
| ≥35 | 1.12 (0.93–1.34) | 0.92 (0.65–1.28) | 1.12 (0.90–1.41) | 1.14 (0.60–2.18) |
| **Parity** | | | | |
| Nulliparous | 1 (reference) | 1 (reference) | 1 (reference) | 1 (reference) |
| ≥1 | 1.24 (1.09–1.41) | 1.48 (1.16–1.89) | 1.18 (1.01–1.39) | 1.16 (0.72–1.85) |
| Not known/missing | 0.49 (0.37–0.65) | 0.99 (0.67–1.46) | 0.37 (0.23–0.61) | 1.16 (0.52–2.56) |
| **Multiple pregnancy** | | | | |
| No | 1 (reference) | 1 (reference) | 1 (reference) | 1 (reference) |
| Yes | 2.48 (2.03–3.04) | 2.57 (1.80–3.65) | 2.35 (1.83–3.04) | 1.40 (0.64–3.08) |
| **Maternal smoking** | | | | |
| Non-smoker/ex-smoker | 1 (reference) | 1 (reference) | 1 (reference) | 1 (reference) |
| Smoker | 1.24 (1.07–1.44) | 1.52 (1.15–2.01) | 1.11 (0.92–1.33) | 1.91 (1.12–3.25) |
| Not known/missing | 0.47 (0.40–0.55) | 0.66 (0.52–0.85) | 0.57 (0.46–0.70) | 0.85 (0.51–1.43) |
| **Anomalies in previous pregnancies** | | | | |
| No | 1 (reference) | 1 (reference) | 1 (reference) | 1 (reference) |
| Yes | 1.16 (0.95–1.41) | 1.24 (0.86–1.79) | 1.25 (0.98–1.59) | 0.56 (0.20–1.56) |
| Not known/missing | 0.46 (0.39–0.53) | 0.70 (0.55–0.89) | 0.53 (0.42–0.66) | 0.96 (0.60–1.53) |
| **Infant sex*** | | | | |
| Male | 1 (reference) | 1 (reference) | 1 (reference) | 1 (reference) |
| Female | 1.29 (1.14–1.46) | 1.08 (0.86–1.35) | 1.33 (1.14–1.55) | 0.79 (0.51–1.22) |
| **Birth weight (g)†** | | | | |
| ≥2500 | 1 (reference) | 1 (reference) | 1 (reference) | 1 (reference) |
| <2500 (low birth weight) | 5.49 (4.84–6.22) | 5.32 (4.26–6.64) | 4.54 (3.88–5.30) | 3.34 (2.13–5.23) |
| **Gestational age at birth (weeks)*†** | | | | |
| ≥ 37$^{+0}$ (term) | 1 (reference) | 1 (reference) | 1 (reference) | 1 (reference) |
| < 37$^{+0}$ (preterm) | 4.53 (4.00–5.14) | 4.72 (3.78–5.90) | 3.84 (3.29–4.49) | 2.76 (1.75–4.35) |
| **CHD anomaly severity** | | | | |
| Less severity | n/a | n/a | n/a | 1 (reference) |
| Moderate severity | | | | 10.1 (4.9–21.1) |
| Most severity | | | | 90.7 (43.4–189) |

Continued

**Table 3** Continued

| Factors | All anomalies | Isolated anomalies | Multiple anomalies | Cardiovascular anomalies |
|---|---|---|---|---|
| Surgery | | | | |
| Not performed or required in the first year after birth | 1 (reference) | 1 (reference) | 1 (reference) | 1 (reference) |
| Performed (or expected) in the first year after birth | 0.86 (0.73–1.01) | 0.80 (0.58–1.10) | 0.68 (0.56–0.82) | 2.93 (1.66–5.17) |
| Not known/missing | 0.83 (0.72–0.95) | 0.92 (0.73–1.17) | 0.76 (0.64–0.91) | 0.75 (0.44–1.26) |

*A priori variable.
†Gestational age at birth and birth weight are strongly correlated (r≥0.7).
CHD, congenital heart defects ; n/a, not applicable; uORs, unadjusted ORs.

the cohort, not to have received surgery or have mothers who were of non-white ethnicity and smokers. Preterm birth was the strongest risk factor for excess infant deaths across all subgroups of congenital anomalies, but the effects of other factors on excess infant deaths varied according to the anomaly subgroup.

Previous studies have shown that significant inequalities in child health and IMR exist between ethnic groups in England and Wales, in which ethnic minority groups generally experience worse outcomes compared with white ethnic groups; however, there is significant heterogeneity within most ethnic minority groups.[14][15] The excess deaths from congenital anomalies in minority ethnic groups are likely to be complex and may be due to an interplay of factors resulting in unequal access to and uptake of antenatal screening and medical and surgical interventions, different attitudes toward congenital anomalies and termination of pregnancy, consanguinity (eg, a risk factor associated with more lethal anomalies),[16] as well as difference in genetics (eg, incidence of genotype mutations), culture (eg, attitude to healthcare and interventions) and behaviour (eg, maternal smoking) between ethnic groups.[4][5][17–19]

The relationship between parity and infant mortality has been well documented, although the exact mechanism is not clear; it is thought that biological, including the impact of maternal age and sociological factors, as well as factors involving in accessing the health services may play a role.[20–23] Smoking during pregnancy is a well-known risk factor for intrauterine growth restriction, prematurity and fetal death. Previous studies have suggested that placental dysfunction and/or abruption lead to fetal hypoxia due to nicotine-induced vasoconstriction during the perinatal period.[24–27] In addition, smoking has been shown to be associated with an increased risk of sudden infant death syndrome.[28] It is possible that these effects are more serious for infants with an underlying anomaly. However, the literature on smoking in relation to congenital anomalies is not unequivocal. The study of Child Death Outcome Panel data in Bradford showed that smoking in pregnancy appeared protective against infant deaths from congenital anomalies, although the study population was small.[29]

Girls with a congenital anomaly had an increased risk of infant death compared with boys with anomalies overall, which is a finding not described previously. However,

**Table 4** aORs for factors associated with infant death by anomaly subgroup

| Factors | All anomalies | Isolated anomalies | Multiple anomalies | Cardiovascular anomalies |
|---|---|---|---|---|
| Ethnicity: other vs white | 2.25 (1.77–2.86) | 1.87 (1.16–3.02) | 2.38 (1.79–3.16) | 1.08 (0.29–4.04) |
| Parity ≥1 vs nulliparous | 1.24 (1.08–1.41) | 1.48 (1.16–1.89) | 1.19 (1.02–1.40) | |
| Maternal smoking: smoker vs non-smoker/ex-smoker | 1.20 (1.02–1.40) | 1.44 (1.07–1.92) | | |
| Infant sex: female vs male | 1.28 (1.13–1.46) | 1.06 (0.85–1.33) | 1.37 (1.17–1.61) | 1.26 (0.78–2.06) |
| Gestational age at birth: preterm vs term | 4.38 (3.86–4.98) | 4.53 (3.62–5.67) | 3.86 (3.30–4.53) | 3.67 (2.19–6.16) |
| CHD severity: moderate vs less Most vs less | n/a | n/a | n/a | 18.4 (8.10–41.7) 229 (90.8–579) |
| Surgery: yes vs no | 0.80 (0.68–0.95) | | 0.71 (0.58–0.86) | 0.34 (0.15–0.75) |

Statistical significance at p<0.05. aORs with 95% CIs are given. Empty cells represent variable that was not included in the multivariable analysis.
aORs, adjusted ORs ; CHD, congenital heart defects; n/a, not applicable.

this is likely to be partially due a male excess of conditions such as pyloric stenosis and potentially hypospadias (not examined in this study), which are sex-specific and rarely lethal (online supplemental table 4). Infants who are born preterm have an increased risk of comorbidities and related complications such as intraventricular haemorrhage, chronic lung disease of prematurity and necrotising enterocolitis,[30] and hence poorer prognostic outcomes in infancy in general compared with infants born at term. Having a major anomaly also increases the chance of an infant being born preterm[30 31]; potentially, both risk factors may contribute to the same chain of event leading to infant death.

The strong association found between infant mortality risk and severity of cardiovascular anomalies is to be expected. However, classification of disease severity among other anomaly subgroups are less well established, as the aetiology of many congenital anomalies is not known.[32] Advances in diagnostic, surgical and medical interventions have generally improved the survival of infants with congenital anomalies who require surgery; the majority of formally lethal anomalies can now be successfully treated.[33] Particularly, advances in cardiac surgery, imaging, prenatal screening and diagnosis have significantly reduced the risk of death for infants with a CHD over time.[34] However, death rates remain high for many severe anomalies in spite of medical and surgical interventions, for example, infants with the most severe cardiac conditions continue to experience a significant risk of postoperative cardiovascular sequelae and other complications.[35] We did not find an association between surgery and isolated anomalies, but it is possible that the prognosis of many less severe isolated anomalies is good regardless of surgery compared with multiple anomalies; however, further subgroup analysis of specific anomaly subtypes is needed to confirm this.

While our results were drawn from data from the period between 1998 and 2017, it is important to note that public health interventions such as the smoking cessation programme in Wales from 2007, and the Antenatal Screening Wales in 2003 may have had an impact on the mortality estimates, although assessing the impact of these programmes is outside the scope of this study. In our study, we found that the infant death rate tended to decrease over the study period for infants with any congenital anomalies.

The main strengths include the national study population and robust study design. The study cohort was identified from a high-quality, population-based congenital anomaly registry with an active surveillance system, which covers all births in Wales. The multiple source reporting system maximises case finding and thus internal validity.[36] A further strength relates to the broad definition of congenital anomalies, which includes structural and chromosomal defects, as well as rare diseases of congenital origin. Consequently, we were able to generate robust epidemiological findings and assess the full impact of a wide group of congenital anomalies and rare diseases on infant death.

As the occurrence of anomaly related infant deaths is rare, it was not possible to investigate risk factors for infant mortality for specific anomalies except for cardiovascular anomaly subgroup. Consequently, heterogeneity inevitably exists within anomaly groups in terms of infant mortality risk. In addition, the number of variables explored in this study was restricted by the data available, which is a common limitation of using routinely collected data. The extent of missing data in variables such as maternal ethnicity and smoking is significant, and self-reporting of smoking is not always reliable[37]; therefore, caution should be taken when interpreting these results. The ORs of anomaly severity associated with death of infants with a CHD have wide CIs due to the small sample size; however, it was not appropriate to group the most severe and moderately severe CHD together, as this would lead to considerable heterogeneity. In addition, as the Welsh population is predominantly white (>95%), a more nuanced analysis by ethnic subgroups is not possible due to small numbers. Finally, some severe cardiac anomalies are not amenable to surgery, or parents do not wish to put their child through years of major surgical intervention and may instead opt for palliative care, thus potential selection bias cannot be excluded even though surgical intervention has improved over years.

## CONCLUSIONS

Congenital anomalies are a leading cause of infant death. Socio-demographic, maternal, infant and interventional factors have a significant impact on death in infants with congenital anomalies by likely different potential mechanisms. Improving access to prenatal care, optimising care for preterm infants, smoking cessation advice and surgery may help lower the risk of infant death.

**Acknowledgements** We thank the Secure Anonymised Information Linkage team for providing the datasets and their assistance with data linkage for this study, and the Congenital Anomaly Register and Information Service (CARIS) team for their support with the CARIS data.

**Collaborators** We thank the Secure Anonymised Information Linkage team for providing the datasets, their assistance with data linkage for this study and conducting all statistical disclosure control and checks; and the Congenital Anomaly Register and Information Service (CARIS) for Wales team for their support with the CARIS data.

**Contributors** PH contributed to the design of the study; acquisition, analysis and interpretation of data; drafting and revising it critically; approved the final version to be published; agrees to be accountable for all aspects of the work in ensuring that questions related to the accuracy or integrity of any part of the work are appropriately investigated and resolved; and accepts full responsibility for the work, overall content and conduct of the study as the guarantor. MQ and JK contributed to the design of the study; acquisition, analysis and interpretation of data for the work; supervised the project and revised it critically; approved the final version to be published; and agree to be accountable for all aspects of the work in ensuring that questions related to the accuracy or integrity of any part of the work are appropriately investigated and resolved. DFT contributed to the acquisition and interpretation of data for the study, revised it critically, approved the final version to be published and agrees to be accountable for all aspects of the work in ensuring that questions related to the accuracy or integrity of any part of the work are appropriately investigated and resolved.

**Funding** This study is partly funded by the National Institute for Health Research (NIHR) Policy Research Programme, conducted through the NIHR Policy Research

Unit in Maternal Health and Care (grant number: 108/0001). The views expressed are those of the author(s) and not necessarily those of the NIHR or the Department of Health and Social Care. PH is funded by a Clarendon – Green Templeton College – Nuffield Department of Population Health Scholarship, University of Oxford.

**Competing interests**  No competing interests.

**Patient consent for publication**  Not applicable.

**Ethics approval**  This study does not involve human participants.

**Provenance and peer review**  Not commissioned; externally peer reviewed.

**Data availability statement**  All data relevant to the study are included in the article or uploaded as supplementary information.

**ORCID iD**
Peter S Y Ho http://orcid.org/0000-0003-4962-6808

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
