## [Reviewer comments · BMJ Paediatrics Open]

This paper was submitted to a another journal from Archives of Disease in Childhood but declined for publication following peer review. The authors addressed the reviewers' comments and submitted the revised paper to BMJ Paediatrics Open. The paper was subsequently accepted for publication at BMJ Paediatrics Open.

ARTICLE DETAILS

TITLE (PROVISIONAL)	RISK FACTORS FOR DEATH IN WELSH INFANTS WITH A CONGENITAL ANOMALY.
AUTHORS	Ho, Peter Quigley, Maria Tucker, David Kurinczuk, Jenny

VERSION 1 – REVIEW

REVIEWER	Reviewer name: Dr. ROGER Charles PARSLow Institution and Country: UNIVERSITY OF LEEDS, PAEDIATRIC EPIDEMIOLOGY GROUP Competing interests: None
REVIEW RETURNED	10-Mar-2021

GENERAL COMMENTS	This analysis of congenital anomaly register data linked with ONS mortality data addresses an important issue in ascertaining the burden of mortality associated with CA and possible areas for improvement in health and social intervention. My comments are aimed at improving the manuscript and addressing some questions about methodology and interpretation. METHODS: 1) The authors need to identify data sources for ethnicity, smoking, medical history on previous anomalies and infant surgery: are they routinely reported to CARIS? Has there been any assessment of their validity? 2) How has the severity of cardiovascular anomalies been classified? Table 3 refers to three levels. The classification should be described in the text. 3) Please supply point estimates and 95% CI for all variables, even if NS. Non-significant results are still valid, and would be used in any meta-analysis. Also, to aid the reader, please give the reference category first in the tables. 4) The description of the sensitivity analyses needs to be more detailed. The authors say they used complete case analysis and imputation, but they also mention that where there was high missingness this was associated with other variables, i.e. not completely missing at random. I would suggest that they describe what they have done more fully. For example, imputation is not appropriate if data are MNAR. This is important as there are significant OR in missing/not known categories. I suggest the authors look at the following paper from Bristol: ****Accounting for missing data in statistical analyses: multiple imputation is not always the answer.
--

	Rachael A Hughes, Jon Heron, Jonathan A C Sterne, Kate Tilling International Journal of Epidemiology, Volume 48, Issue 4, August 2019, Pages 1294–1304, https://doi.org/10.1093/ije/dyz032**** 5) You will note that this paper refers to the use of causal diagrams and this approach would be valuable in determining how the data is modelled and also aid in its interpretation. It may also remove the need to use stepwise elimination of variables - there is divided opinion on this method but I would prefer to see a more sophisticated causal diagram approach. INTERPRETATION and DISCUSSION The IMRs and case fatality data are quite straightforward and a valuable addition to the field. The interpretation of risk factors where there is a lot of missing data in some variables and a lack of clarity in how some factors are related is difficult. If the authors used a causal diagram approach and looked at the reasons for missing data in more detail, I think their conclusions could be refined. For example, the findings of excess mortality in smokers is not clear cut to me: there are a lot of missing data (over 40%), and reliability of self-report of smoking in pregnancy is not good: see this report from Scotland: ****Reliability of self reported smoking status by pregnant women for estimating smoking prevalence: a retrospective, cross sectional study. BMJ 2009; 339 doi: https://doi.org/10.1136/bmj.b4347**** Also, the literature on smoking in relation to CA is not unequivocal: In a small study of Child Death Outcome Panel data in Bradford, Catriona Firth and colleagues, found that children who died from CA were more likely to have been born at term and that smoking in pregnancy was protective. ****Infant deaths from congenital anomalies: novel use of Child Death Overview Panel data. ADC November 2018 - Volume 103 – 11; http://dx.doi.org/10.1136/archdischild-2017-314256****. I am not sure how to interpret this in the context of an almost bi-ethnic population where very few woman from the Pakistani population smoke, but have high rates of CA. I am not sure that the authors will be able to provide interpretation either, but should at least address the difficulties these results present. The relationship between ethnicity and socioeconomic status is also very difficult to disentangle. One way of looking at this would be to examine the interactions between these related variables. At least the difficulties in interpretation need highlighting. This might help in making conclusions and identifying the 'likely different potential mechanisms' mentioned in the conclusions.
--	--

REVIEWER	Reviewer name: Dr. Sam J Oddie Institution and Country: Bradford Royal Infirmary, Bradford Neonatology Competing interests: None
REVIEW RETURNED	24-Mar-2021

GENERAL COMMENTS	I enjoyed reading this paper, and in general I found the methodology to be faultless, and the presentation appropriately brief and clear. I had some minor points in which I thought the paper could be
--

	improved. I thought the description of female as reference for mortality comparisons by gender was confusing in table 3. The finding on gender is the most striking, and more clarity on whether this is wholly or partly explained by the debatable classification of pyloric stenosis as an anomaly seems like a good idea - I think the broad message here is a little confused. I note the difference in mortality for downs - and consider this a major finding. Is a consideration of multiple testing an issue that could be rebutted? I think it would be interesting to consider PAR. by this I mean, what would the risk of death be if it was as low for all deprivation categories (and perhaps other risks??) as it is for the least deprived group? If valid and possible, such a calculation would add interest to the paper. I note that the explanation of possible mechanisms by which ethnicity might alter the risk of death does not include consanguinity. This is associated with more lethal anomaly, and is likely a major factor. I think the authors should consider a more granular description of ethnicity. Pakistani ethnicity alone is common, and strongly associated with consanguinity, which is an important potential mechanism here. In table 1 I am concerned that the cells for isolated and multiple anomalies do not appear to sum to the total. perhaps this is easily explained, but more clarity might help.
--	---

REVIEWER	Reviewer name: Ms. Deborah Ridout Institution and Country: Institute of Child Health, Paediatric Epidemiology Biostatistics Competing interests: None
REVIEW RETURNED	29-Mar-2021

GENERAL COMMENTS	This is an interesting study which explores and quantifies the association between pre-specified risk factors and mortality of infants with a congenital anomaly. I feel the authors need to clarify some of their methods and provide further detail and justification before it is suitable for publication. Specific comments  • The title should reflect this study refers to Wales only. Also, the definition of mortality should be stated more explicitly in the methods, as within 1 year of life. • Figure 1 is very helpful although it is important that full details are provided, including where information is missing, so that all cases are accounted for and numbers match up. Also, n (%) should be specified for all categories. • All pre-defined risk factors and the levels of categorisation should be stated explicitly in the methods. • Stepwise univariate analysis has been conducted and only those factors that reached a statistical significance, based on an arbitrary cut-point have been included in the multivariable models. I would not recommend this approach and would advise including all important variables and observe and report the size of the effect based on clinical importance. • Multiple imputation has been conducted to account for missing data, however no details are provided. Full details should be included. Also how was ethnicity imputed, how well did the imputation work here? • How was the stepwise regression implemented with the multiple imputed data – how was the 'best' model selected for each of the
---

	imputed data sets.  • The column headers in table 2 should be changed as currently it is misleading. Currently column 2 could infer death rate. I don't understand the footnote to this table, particularly the reference to P values. What comparison is this referring to, there is no mention of this in the methods section. Also why is there reference to the regression model here – this seems misplaced. • Tables 3&4 – NS is not appropriate – all results, including statistically non-significant ones should be reported.
--	---

VERSION 1 – AUTHOR RESPONSE

28th June, 2021.

Dear Editor-in-chief, BMJ Paediatrics Open,

In response to the recommendation from the editor of Archives of Disease in Childhood, we have reviewed our paper (please see the letter in response to the peers review- responses/reviews are highlighted in red in the manuscript) and would like to submit a revised manuscript entitled "RISK FACTORS FOR DEATH IN WELSH INFANTS WITH A CONGENITAL ANOMALY" for publication consideration in BMJ Paediatrics Open.

Congenital anomalies are a leading cause of infant mortality. Population-based studies on risk factors contributing to death of infants with congenital anomalies are limited. In addition, most of the existing studies focused on a few major anomaly subgroups such as neural tube defects, and certain cardiovascular and digestive system anomalies. This cohort study used linked de-identified data from the Welsh Congenital Anomaly Register and Information Service (CARIS) and livebirths and deaths from the Office for National Statistics (ONS) to investigate risk factors for death of infants born with congenital anomalies and rare diseases of congenital origin as defined by CARIS. This study has found that almost a third of all infant deaths had an associated anomaly. Preterm birth is the strongest factor significantly associated with excess infant deaths in all anomaly subgroups investigated; the effects of maternal ethnicity, maternal smoking, parity, infant sex, anomaly severity and surgery on excess infant case fatality vary according to the anomaly subgroup. Therefore, improving access to prenatal care, smoking cessation advice, optimising care for preterm infants, and surgery may help lower the risk of infant death.

We trust the findings of this study help contribute to the research gap in this field.

Many thanks for your consideration.

Sincerely,

Dr. P Ho

National Perinatal Epidemiology Unit,
Nuffield Department of Population Health,
University of Oxford

VERSION 2 – REVIEW

REVIEWER	Reviewer name: Dr. Emmanouil Bagkeris Institution and Country: Imperial College London, National Heart and Lung Institute / Genomic and Environmental Medicine Competing interests: None
REVIEW RETURNED	17-Jul-2021

GENERAL COMMENTS	This is a very interesting and well written manuscript. Below are some recommendations and suggestions:  1. Why did you not consider a time to event analysis? Date of death should be routinely reported. 2. The statistical analysis section is confusing regarding the way confounders were selected. One sentence suggests that the confounders were selected a priori based on the literature and the second suggests that variables associated with death ($p < 0.1$) were considered in the multivariable models. Can you please describe the confounding factor selection strategy better? 3. Perhaps instead of figure 2, explain the choice of the confounding factors using a directed acyclic diagram (D.A.G.) 4. Although the discussion section suggests that gestational age and major anomalies may synergistically contribute to child mortality and although both factors are available in this study, the synergistic effect was not explored in this manuscript. Would you consider exploring this association by exploring the interaction term of gestational age and major anomalies or by stratifying the analyses for term and preterm children. If you choose to do the later, it would be interesting to see whether the estimates of the confounding factors suggest a stronger association with mortality among those born preterm vs. term. 5. In the last paragraph of the discussion you state that interventions such as antenatal screening and smoking cessation have been implemented over the study period and may have an impact on the child mortality. If you have such concerns why was calendar year not considered in the analyses? 6. Table 3 and table 4 have discrepant variable labelling for Congenital heart defects (CHD) and CVS--which is not explained in abbreviations. The factor congenital heart defects has very wide confidence intervals in table 3 and 4. Have you considered regrouping the moderate and most severity together? The very wide CIs should be due to the very small number of those in the "Most severity" group.
---

REVIEWER	Reviewer name: Dr. Archana Patel Institution and Country: Research, 9/11 Vasant Nagar, Nagpur, Maharashtra, 440022, India Competing interests: None
REVIEW RETURNED	03-Sep-2021

GENERAL COMMENTS	Congenital anomalies(CA) are the second leading cause of infant death in the UK. Among those born with CA (structural, chromosomal or metabolic abnormalities), the authors aim to investigate the risk factors for infant deaths. This is a registry data based cohort that included livebirths born from between 1998 and 2016 that were included both in the Congenital Anomaly Registry and the death registry of the ONS. The authors did not provide the figure that illustrates the flow of data of the cohort of all live births leading up to the analysed population in this study. It should include the total number of livebirths during that period, the number that were reported in CARIS and the numbers that were not in CARIS, the number(%) of
---

	infants in CARIS that were alive/dead and the numbers(%) not in CARIS that were alive/dead at end of 12 months. Finally information regarding congenital anomalies(confirmed and probable) in those that were dead in either (CARIS and Non CARIS) as the information of CA may be available in the ONS. The information provided in Table 1 can be revised accordingly and included in this flow diagram. If there are more CA identified in the Non CARIS data base and their mortality outcomes at the end of 12 months, the IMR in children with CA may change. The EUROCAT provided a list of 95 variables for the registry. Can the authors explain why only 12 variables were included in this analysis ? What variables are available in CARIS ? Page 5, line 13 – “(2) All confirmed and probable cases of congenital anomalies”. This is already included in (1) , so what does this statement mean ? Page 5 – “Infants were considered as having multiple anomalies (or diseases) if more than one anomaly (or diseases) was diagnosed, either within the same body system or involving different body systems.” Please provide an example. If an infant has an eye and ear abnormality – is that isolated or multiple ? If an infant has a syndrome with a renal and cardiac anomaly, were they double counted in both syndromic and renal system category ?? The supplementary table reports the frequencies of different types of anomalies. The intent of identifying risk factors of CA mortality is to be able to either modify the risk factors or to provide targeted interventions and careful monitoring of those at higher risk of mortality. Infants can have a isolated anomaly but of a serious kind – eg Esophageal atresia Or not a serious kind i.e a cleft lip. Both can be isolated. But one is incompatible with life and the other will have a low risk of dying. Therefore for improving care to those at high risk of death, this sort of categorization of isolated, multiple is not helpful. Syndromic is fine. It would be better to categorize the CAs in terms of clinical severity – Minor, Intermediate, Major, instead of isolate, multiple, syndromic and cardiovascular as these do not guide clinical follow up or preventive measures. This manuscript therefore needs major revisions, analysis and a resubmission. It needs to include data on CA available at death of the infant that was not otherwise included in CARIS, explain what predictor variables are available in CARIS as compared to EUROCAT and the reasons for including or excluding the variables, and, re-categorization of the CAs according to severity to proceed to further analysis of CA specific IMR, category specific IMR and their risk factors.
--	---

VERSION 2 – AUTHOR RESPONSE

5.11.2021

Dear Editor, BMJ Paediatric Open,

Thanks for the reviews and comments. Please find the responses below and in the manuscript highlighted in red.

Peter

Editor(s)' Comments to Author (if any):

Associate Editor

Comments to the Author:

Respond in full to the reviewers

Discussion delete the sentence "This is the first UK population-based cohort study of which we are aware investigating the impact of overall and selected groups of congenital anomalies and rare diseases on infant death." journal style is to avoid describing a study as the first.

What this study adds needs rewriting so that it is shorter and more focused, eg " Preterm birth was the strongest risk factor for excess infant deaths" to replace the first sentence. may I suggest the statements below

"A third of infant deaths in Wales involved an infant with a congenital anomaly"

"Socioeconomic factors such as ethnicity and smoking are risk factors"

Discussion would benefit from discussion on inequalities in child health

Response: Revised in respective sections.

Reviewer: 1

Dr. Emmanouil Bagkeris, Imperial College London

Comments to the Author

This is a very interesting and well written manuscript. Below are some recommendations and suggestions:

1. Why did you not consider a time to event analysis? Date of death should be routinely reported.

Response: The vast majority of infant deaths occur in the first few weeks and therefore it is common to treat infant mortality as a binary variable rather than a time to event analysis. In our study, only the week of birth (rather than the exact date) was available, and therefore a time to event analysis based on weeks would not have been appropriate All study babies were followed up from birth and our outcome of interest was whether these babies died or survived at 1 year of age as a binary outcome, hence logistic regression analysis was appropriate.

2. The statistical analysis section is confusing regarding the way confounders were selected. One sentence suggests that the confounders were selected a priori based on the literature and the second suggests that variables associated with death ($p < 0.1$) were considered in the multivariable models. Can you please describe the confounding factor selection strategy better?

Response: revised in Methods section.

3. Perhaps instead of figure 2, explain the choice of the confounding factors using a directed acyclic diagram (D.A.G.)

Response: revised in Figure 2.

4. Although the discussion section suggests that gestational age and major anomalies may synergistically contribute to child mortality and although both factors are available in this study, the synergistic effect was not explored in this manuscript. Would you consider exploring this association by exploring the interaction term of gestational age and major anomalies or by stratifying the analyses for term and preterm children. If you choose to do the later, it would be interesting to see whether the estimates of the confounding factors suggest a stronger association with mortality among those born preterm vs. term.

Response: Revised in Methods. Interaction term was explored. There was no interaction between gestational age and other variables in the final model, hence no further stratification analysis for term vs. preterm infant was conducted.

5. In the last paragraph of the discussion you state that interventions such as antenatal screening and smoking cessation have been implemented over the study period and may have an impact on the child mortality. If you have such concerns why was calendar year not considered in the analyses?

Response: We have now included year of birth in the final model as a sensitivity analysis (Supplementary Table 3). The effects of each variable did not change substantially when they were adjusted for year of birth. We found that the adjusted effect of calendar year (as a continuous variable) on infant deaths was mild.

6. Table 3 and table 4 have discrepant variable labelling for Congenital heart defects (CHD) and CVS-- which is not explained in abbreviations. The factor congenital heart defects has very wide confidence intervals in table 3 and 4. Have you considered regrouping the moderate and most severity together? The very wide CIs should be due to the very small number of those in the "Most severity" group.

Response: We have now clarified the abbreviations. We considered combining the moderate and severe groups for CHD. However, the prognosis for the most severe CHD such as hypoplastic heart syndromes are significantly worse than those for moderate severe CHD. Hence we do not think it is appropriate to group the most severe and moderately severe CHD together, as this would lead to considerable heterogeneity. However, we have interpreted the results cautiously and noted the wide CI in the revised manuscript.

Reviewer: 2

Dr. Archana Patel

Comments to the Author

Congenital anomalies(CA) are the second leading cause of infant death in the UK. Among those born with CA (structural, chromosomal or metabolic abnormalities), the authors aim to investigate the risk factors for infant deaths.

This is a registry data based cohort that included livebirths born from between 1998 and 2016 that were included both in the Congenital Anomaly Registry and the death registry of the ONS.

The authors did not provide the figure that illustrates the flow of data of the cohort of all live births leading up to the analysed population in this study. It should include the total number of livebirths during that period, the number that were reported in CARIS and the numbers that were not in CARIS, the number(%) of infants in CARIS that were alive/dead and the numbers(%) not in CARIS that were alive/dead at end of 12 months. Finally information regarding congenital anomalies(confirmed and probable) in those that were dead in either (CARIS and Non CARIS) as the information of CA may be available in the ONS. The information provided in Table 1 can be revised accordingly and included in this flow diagram. If there are more CA identified in the Non CARIS data base and their mortality outcomes at the end of 12 months, the IMR in children with CA may change.

Response: Revised in Table 1 and Figure 1.

The EUROCAT provided a list of 95 variables for the registry. Can the authors explain why only 12 variables were included in this analysis ? What variables are available in CARIS ?

Response: The choice of socio-demographic, maternal, infant and interventional variables were based on a literature review, clinical importance, as well as data quality issues, for example, missingness of the variables. For example, variables such as maternal medical history and therapies have much missing data; hence they were not appropriate to include as this would introduce bias.

Page 5, line 13 – "(2) All confirmed and probable cases of congenital anomalies". This is already included in (1) , so what does this statement mean ?

Response: Revised in Methods.

Page 5 – "Infants were considered as having multiple anomalies (or diseases) if more than one anomaly (or diseases) was diagnosed, either within the same body system or involving different body systems."

Please provide an example. If an infant has an eye and ear abnormality – is that isolated or multiple ?

Response: Revised in Methods section.

If an infant has a syndrome with a renal and cardiac anomaly, were they double counted in both syndromic and renal system category ??

Response: Revised in Methods section.

The supplementary table reports the frequencies of different types of anomalies. The intent of identifying risk factors of CA mortality is to be able to either modify the risk factors or to provide targeted interventions and careful monitoring of those at higher risk of mortality.

Infants can have a isolated anomaly but of a serious kind – eg Esophageal atresia Or not a serious kind i.e a cleft lip. Both can be isolated. But one is incompatible with life and the other will have a low risk of dying. Therefore for improving care to those at high risk of death, this sort of categorization of isolated, multiple is not helpful. Syndromic is fine. It would be better to categorize the CAs in terms of clinical severity – Minor, Intermediate, Major, instead of isolate, multiple, syndromic and cardiovascular as these do not guide clinical follow up or preventive measures.

Response: This study used the EUROCAT classification of congenital anomalies, the results of which can be compared with other studies with a similar methodology for anomaly classification. Apart from congenital heart defects, which have severity classified by EUROCAT, there is no known agreed system to objectively classify other anomalies based on clinical severity.

This manuscript therefore needs major revisions, analysis and a resubmission. It needs to include data on CA available at death of the infant that was not otherwise included in CARIS, explain what predictor variables are available in CARIS as compared to EUROCAT and the reasons for including or excluding the variables, and, re-categorization of the CAs according to severity to proceed to further analysis of CA specific IMR, category specific IMR and their risk factors.

VERSION 3 – REVIEW

REVIEWER	Reviewer name: Dr. Emmanouil Bagkeris Institution and Country: Imperial College London, National Heart and Lung Institute / Genomic and Environmental Medicine Competing interests: None
REVIEW RETURNED	30-Nov-2021
GENERAL COMMENTS	The authors have adequately addressed the points raised in the first review. Perhaps they may want to highlight the consistent protective effect of having surgery on infant death in the box "What this study adds".

VERSION 3 – AUTHOR RESPONSE